# Interactions of Oxysterols with Atherosclerosis Biomarkers in Subjects with Moderate Hypercholesterolemia and Effects of a Nutraceutical Combination (*Bifidobacterium longum* BB536, Red Yeast Rice Extract) (Randomized, Double-Blind, Placebo-Controlled Study)

**DOI:** 10.3390/nu13020427

**Published:** 2021-01-28

**Authors:** Stefania Cicolari, Chiara Pavanello, Elena Olmastroni, Marina Del Puppo, Marco Bertolotti, Giuliana Mombelli, Alberico L. Catapano, Laura Calabresi, Paolo Magni

**Affiliations:** 1Dipartimento di Scienze Farmacologiche e Biomolecolari, Università degli Studi di Milano, 20133 Milan, Italy; stefania.cicolari@unimi.it (S.C.); chiara.pavanello@unimi.it (C.P.); alberico.catapano@unimi.it (A.L.C.); laura.calabresi@unimi.it (L.C.); 2Centro E. Grossi Paoletti, Dipartimento di Scienze Farmacologiche e Biomolecolari, Università degli Studi di Milano, 20133 Milan, Italy; 3Servizio di Epidemiologia e Farmacologia Preventiva (SEFAP), Dipartimento di Scienze Farmacologiche e Biomolecolari, Università degli Studi di Milano, 20133 Milan, Italy; elena.olmastroni@unimi.it; 4Dipartimento di Medicina e Chirurgia, Università degli Studi di Milano Bicocca, 20900 Monza, Italy; marina.delpuppo@unimib.it; 5Dipartimento di Scienze Biomediche, Metaboliche e Neuroscienze, Università degli Studi di Modena e Reggio Emilia, 41126 Modena, Italy; marco.bertolotti@unimore.it; 6Centro Dislipidemie, ASST Grande Ospedale Metropolitano Niguarda, 20162 Milan, Italy; giuliana.mombelli@ospedaleniguarda.it; 7IRCCS MultiMedica, Sesto S. Giovanni, 20099 Milan, Italy

**Keywords:** oxysterols, 24-OHC, 25-OHC, 27-OHC, cholesterol metabolism, probiotic, cardiovascular risk, hypercholesterolemia, monacolin K, nutraceutical

## Abstract

Background: Oxysterol relationship with cardiovascular (CV) risk factors is poorly explored, especially in moderately hypercholesterolaemic subjects. Moreover, the impact of nutraceuticals controlling hypercholesterolaemia on plasma levels of 24-, 25- and 27-hydroxycholesterol (24-OHC, 25-OHC, 27-OHC) is unknown. Methods: Subjects (*n* = 33; 18–70 years) with moderate hypercholesterolaemia (low-density lipoprotein cholesterol (LDL-C:): 130–200 mg/dL), in primary CV prevention as well as low CV risk were studied cross-sectionally. Moreover, they were evaluated after treatment with a nutraceutical combination (*Bifidobacterium longum* BB536, red yeast rice extract (10 mg/dose monacolin K)), following a double-blind, randomized, placebo-controlled design. We evaluated 24-OHC, 25-OHC and 27-OHC levels by gas chromatography/mass spectrometry analysis. Results: 24-OHC and 25-OHC were significantly correlated, 24-OHC was correlated with apoB. 27-OHC and 27-OHC/total cholesterol (TC) were higher in men (median 209 ng/mL and 77 ng/mg, respectively) vs. women (median 168 ng/mL and 56 ng/mg, respectively); 27-OHC/TC was significantly correlated with abdominal circumference, visceral fat and, negatively, with high-density lipoprotein cholesterol (HDL-C). Triglycerides were significantly correlated with 24-OHC, 25-OHC and 27-OHC and with 24-OHC/TC and 25-OHC/TC. After intervention, 27-OHC levels were significantly reduced by 10.4% in the nutraceutical group Levels of 24-OHC, 24-OHC/TC, 25-OHC, 25-OHC/TC and 27-OHC/TC were unchanged. Conclusions: In this study, conducted in moderate hypercholesterolemic subjects, we observed novel relationships between 24-OHC, 25-OHC and 27-OHC and CV risk biomarkers. In addition, no adverse changes of OHC levels upon nutraceutical treatment were found.

## 1. Introduction

Several lipid biomarkers may contribute to atherosclerosis-related cardiovascular diseases (ASCVD) and, among them, low-density lipoprotein (LDL) is a well-established causal factor [1]. Cholesterol metabolism includes de-esterification in lysosomes to generate free cholesterol which is used for several cellular processes [2] and could also undergo enzymatic oxidation in the mitochondria, leading to the formation of oxysterols [3]. These cholesterol metabolites, precursors of bile acids, are involved both in physiological mechanisms, interdependent with lipid and glucose metabolism, as well as in biological functions such as immune and cerebral homeostasis [4,5]. Additionally, oxysterols have been related to some pathological processes (e.g., atherosclerosis, type 2 diabetes mellitus, neurodegenerative disorders, cancer), for which they may represent potential innovative biomarkers [6]. We focused our attention on 24-hydroxycholesterol (24-OHC), 25-hydroxycholesterol (25-OHC) and 27-hydroxycholesterol (27-OHC), which are mainly synthesized by cytochrome P450 family 46 subfamily A member 1 (CYP46A1), cholesterol 25-hydroxylase (CH25H) and cytochrome P450 family 27 subfamily A member 1 (CYP27A1), respectively. Interestingly, these oxysterols have been shown to act as a link between cholesterol metabolism and different physiological systems [4,7]. 

Moderate hypercholesterolemia is frequently observed in subjects with medium/low 10 years CV risk, represents a significant population burden, particularly when combined with unhealthy lifestyle habits [8] and is often underdiagnosed and undertreated, therefore highly contributing significantly to ASCVD prevalence [9]. Few studies have evaluated the circulating levels of 24-, 25- and 27-OHC in subjects with moderate hypercholesterolemia, also after statin treatment [10,11].

Therapeutical strategies for this condition may include the use of low-efficacy/low-dose statins and/or nutraceutics [12,13,14]. This treatment may offer multi-faceted effects and significant advantages over no-treatment or inadequate adherence to drug treatment, sometimes due to adverse effects and other reasons [15,16,17]. To date, information is lacking about the impact of nutraceuticals, targeted to improve the atherogenic lipid profile, on the synthesis of oxysterols, downstream of inhibition of cholesterol biosynthesis and absorption. Based on these considerations, the main objective of the present study was to evaluate the interactions of oxysterols with cardiovascular biomarkers and subsequently to study the effects of a nutraceutical treatment (*Bifidobacterium longum* BB536, RYR extract, niacin, coenzyme Q10) on their circulating levels. This nutraceutical combination was previously found to be quite effective in reducing LDL cholesterol (LDL-C) and total cholesterol (TC) levels in moderately hypercholesterolemic subjects [18]. The study yielded novel data on these biochemical events, with potential health implications and supported the safety profile of this nutraceutical combination. 

## 2. Materials and Methods

### 2.1. Study Design and Population

The study was conducted at the Centro Dislipidemie (ASST Grande Ospedale Metropolitano Niguarda, Milan, Italy) in the period 2015–2017, according to the guidelines of the Declaration of Helsinki. The study cohort included 33 subjects (16 males and 17 females) with moderate hypercholesterolemia, median age: 57 years (Q1 = 48 and Q3 = 63 years) and low total CVD risk at (0%: 8 subjects; 1%: 15; 2%: 5; 3%: 3; 4%: 1; 5%: 1), assessed by the SCORE Risk Charts for low risk countries (like Italy) [19]. Inclusion criteria were subjects in primary CV prevention, age: 18–70 years, non-smokers, LDL-C: 130–200 mg/dL. Exclusion criteria included: pregnancy, smoking (current or previous), diagnosis of diabetes mellitus, chronic liver disease, renal disease, or severe renal impairment treated with insulin or antidiabetic drugs; untreated, uncontrolled or severe arterial hypertension; obesity (body mass index (BMI) >30 kg/m^2^); any pharmacological treatments known to interfere with the study treatment (including statins, ezetimibe, fibrates, thyroid hormones); and patients enrolled in another research study in the past 3 months. The study cohort included 5/32 subjects (15.6%) undergoing drug therapy for arterial hypertension, as reported in [18], together with average food intake according to sex.

The same cohort was also included in a 12-week intervention study (a randomized controlled trial (RCT) design with parallel-groups (NCT02689934)). Subjects were randomly assigned to receive either placebo (1 sachet/d; *n* = 17) or a nutraceutical combination (Lactoflorene Colesterolo^®^-1 sachet/d; granules for oral suspension; with taste/appearance identical to the placebo sachet) composed of 1 bn UFC *Bifidobacterium longum* BB536, RYR extract (10 mg monacolin K), 16 mg niacin, 20 mg coenzyme Q10; *n* = 16) or (Figure 1; CONSORT flow diagram). The randomization table was produced by computer-generated random numbers. The prevalence of subjects with drug-controlled hypertension was 18.8% in the placebo arm and 12.5% in the intervention arm. The study was approved by the Ethics Committee of ASST Grande Ospedale Metropolitano Niguarda. A written informed consent was obtained from all subjects.

### 2.2. Clinical Procedures

Patients underwent a fasting venous blood sampling and a full clinical examination, with determination of height, body weight, waist circumference, heart rate, and systolic and diastolic blood pressure (SBP, DBP). Bioelectric impedance analysis (ViScan device-Tanita Inc., Tokyo, Japan) was used to estimate % abdominal fat mass (BIA (%)) and % visceral fat rating (VFR (%)), according to reported procedures [12]. Plasma samples were immediately separated by centrifugation, and aliquots were immediately stored at −20 °C. In the present analysis, based upon the study reported in [18], we evaluated basal and post-intervention circulating oxysterol levels. CV biomarkers (TC, non-HDL-C, triglycerides (TG), HDL-C, apolipoprotein (apo)AI, apoB, lipoprotein(a) (Lp(a)), proprotein convertase subtilisin/kexin type 9 (PCSK9)) from this study were used for correlation analysis [18]. Data retrieval, analysis, and the preparation of the manuscript were solely the responsibility of the authors.

### 2.3. Immunometric and Biochemical Assays

In all blood samples, TC, HDL-C, TG, apoAI, apoB, Lp(a), fasting plasma glucose (FPG), uric acid were measured according to a standard automated clinical procedure (Cobas system, Roche, Italy). LDL-C was calculated according to the Friedewald formula. Non-HDL-C was calculated as TC minus HDL-C. Enzyme-linked immunosorbent assay (ELISA) kits were used according to manufacturer’s specifications to quantify fibroblast growth factor (FGF) 19, FGF21 and PCSK9 [20] (R&D System, Minneapolis, MN, USA). Oxidized LDL (oxLDL), and insulin were measured by ELISA kits (Mercodia, Sweden). The homeostasis model assessment of insulin resistance (HOMA-IR) index was calculated according to this equation: HOMA-IR = [fasting glucose (mg/dL) × insulin (mUI/L)/405].

### 2.4. Determination of Serum Levels of Oxysterols

Oxysterols were analyzed as previous described [21] using a Thermofinnigan GC-Q instrument supplied with an ion trap source. Oxysterols separation was obtained with an HP5 (Agilent, Lexington, MN, USA) capillary column 0.25 mm i.d., 0.25 μm film thickness, 30 m length, operating at 1 mL/min helium flow rate. Column temperature was programmed from 200 to 300 °C at 20 °C/min. Ions were recorded at *m*/*z* 353 for 19-hydroxycholesterol and *m*/*z* 462 for deuterated 27-OHC (internal standards), *m*/*z* 456 for 27- and 25-OHC and *m*/*z* 413 for 24-OHC. Endogenous hydroxysterol concentrations were calculated with a standard curve, prepared as described [21], and the peak area ratio (sterol/IS) found in the sample. Noteworthy, in the literature the same chemical compound is called both 26-OHC and 27-OHC [22]. This second term over time has become common and to prevent misunderstandings in this study the term “27-OHC” will be used [23].

### 2.5. Statistical Analysis 

Sample size calculation. According to [18], a group sample size of 16 per arm achieves 80% power to detect a difference of 20 mg/mL in absolute changes (12 weeks-0 week) in LDL-C levels (mg/mL), between the null hypothesis that in both arms the means of change in LDL-C are 10 mg/mL and the alternative hypothesis that the mean of change in LDL-C in the treatment arms is −10 mg/mL [12]. The estimated group standard deviations were 25 mg/mL per arm, with a significance level of 5% using a two-sided two-sample *t*-test.

Results are shown as median and interquartile ranges (Q1 and Q3) for all parameters. Correlations between circulating oxysterols as well as oxysterols normalized for TC and several covariates were analyzed using a Pearson correlation coefficient. Oxysterol levels are expressed both as absolute values as well as oxysterol-to-TC ratios, in order to correct for differences in plasma TC concentration and for the evidence that sterols are transported by plasma lipoproteins, in line with the available literature [24]. 

Differences in median values between treatment and placebo arms at baseline were assessed by Wilcoxon-rank sum test. Distributions of percent changes in oxysterols and lipid levels from baseline (0 week) to the end of follow-up (12 weeks treatment) were compared between the placebo and the treatment groups with Wilcoxon rank sum tests. At the end of the follow-up timeframe, the difference in the median percent change observed in the treatment group minus the median percent change in the placebo group at that time was used to summarize the treatment effect. Similar comparisons of percent changes in lipid levels between arms used the same approach. All tests are 2-sided; *p* values 0.05 are considered statistically significant. Statistical analysis was conducted by using both SAS Software version 9.3 (SAS, Cary, NC, USA) and R Software version 3.6.2.

## 3. Results

### 3.1. Study Population

All subjects were in primary CV prevention and showed a moderate hypercholesterolemia. The data analysis was conducted on the 30/33 subjects who also completed the interventional study (Figure 1). The clinical and biochemical data suggest that this cohort showed normal body weight, BMI, waist circumference, BIA and VFR (Table 1). Median TC was 270 (246, 288) mg/dL and LDL-C was 179 (169, 195) mg/dL. TG, HDL-C, insulin sensitivity and blood pressure were in the reference range [25,26]. 

### 3.2. Analysis of Oxysterols in the Study Population 

The 24-OHC values (89 (73, 109) ng/mL) of the study cohort are reported in Table 1. They did not differ according to sex (males: 89.4 (73.5, 110.5) ng/mL; females: 89 (73, 109) ng/mL) (Figure 2A) and did not correlate with age (Table 2). The 24-OHC/TC ratio was 34 (27, 41) ng/mg when considering the entire cohort (Table 1), without sex difference (males 37 (27, 43) ng/mg; females 31 (27, 38)) (Figure 2B) and no correlation with age (Table 2). Intriguingly, 24-OHC was positively correlated with TG (*p* = 0.004) and apoB (*p* = 0.012) and, after normalizing the values for TC, only the correlation with TG was still present (*p* = 0.024) (Table 2). The values of 25-OHC were 84.2 (60.5, 96) ng/mL (Table 1) and did not differ according to sex (males 84.7 (72, 101) ng/mL; females 71.5 (54, 89) ng/mL) (Figure 2A), nor they correlated with age (Table 2). 25-OHC/TC levels (29 ng/mg (25, 38) (Table 1) were not different according to sex (males 32 (26, 41) ng/mg; females 27 (18, 32) ng/mg) (Figure 2B) and did not correlate with age (Table 2). Interestingly, both 25-OHC and 25-OHC/TC showed a significant positive correlation with TG (*p* = 0.007 and *p* = 0.028) (Table 2). Moreover, 24-OHC and 25-OHC levels were significantly correlated (*p* = 0.0002) (Table 2). The 27-OHC values (183.5 ng/mL (152, 211)), shown in Table 1, significantly (*p* = 0.02) diverged between males (209 (173, 230) ng/mL) and females (167.7 (126, 193) ng/mL) (Figure 2A) and did not correlate with age (Table 2). 27-OHC/TC values (72 ng/mg (51, 80)) were also different (*p* = 0.008) according to sex (males 77 (61, 90) ng/mg; females 56 (46, 72) ng/mg (Figure 2B) and did not correlate with age. In addition, 27-OHC showed a positive correlation trend with TG (*p* = 0.056) and non-HDL-C (*p* = 0.065), while it was significantly correlated with creatinine (*p* = 0.017). 27-OHC/TC was negatively correlated with HDL-C (*p* = 0.006) and apoAI (*p* = 0.05), whereas it was positively correlated with abdominal circumference (*p* = 0.023) and VFR (*p* = 0.021) (Table 2). Lp(a) was correlated with 24-OHC/TC (*p* = 0.021), 25-OHC (*p* = 0.045) and 25-OHC/TC (*p* = 0.013). Moreover, PCSK9 levels were negatively correlated with 27-OHC/TC (*p* = 0.013) (Table 2).

### 3.3. Effect of Nutraceutical Treatment on Oxysterols Plasma Levels

The participants of the cross-sectional study were also randomized to either nutraceutical combination or placebo (Figure 1). No differences between arms were found for the different variables, except age and apoB (Table 1). After nutraceutical intervention, compared to placebo, TC was significantly reduced (*p* < 0.0001; −16.7%), together with LDL-C (*p* < 0.0001; −25.7%), as previously reported [18] (please refer to this article for additional data on CVD biomarkers). After the normalization for TC (24-OHC/TC, 25-OHC/TC and 27-OHC/TC), oxysterol levels did not differ between the 2 groups. When considering the absolute value of circulating oxysterols, in the nutraceutical treatment arm 27-OHC concentrations were significantly (*p* = 0.008) decreased (−10.4%), whereas 24-OHC and 25-OHC levels did not change (Table 3).

## 4. Discussion

The present study was aimed at analyzing the relationship between circulating oxysterols, namely 24-, 25- and 27-OHC, with biomarkers related to atherosclerosis in subjects with moderate hypercholesterolemia. We also evaluated the effect of a nutraceutical combination containing *Bifidobacterium longum* BB536 and RYR and aimed to reduce hypercholesterolemia, on the circulating levels of these oxysterols. In our cohort of moderate hypercholesterolemic subjects, 24-OHC values were found to be within the range (33.2–227.0 ng/mL) previously reported for different populations [4]. On the contrary, the values of 25-OHC were almost 3 fold higher compared to those reported (range 2.0–31.0 ng/mL) previously [4]. Together with data indicating that hypercholesterolemic males have significantly higher 25-OHC levels compared to healthy males [10], our findings suggest that this elevation may be peculiar for this condition. 27-OHC values in our population were within the previously reported 27-OHC range (43.6–196.0 ng/mL) [4]. Different measurement techniques (high-performance liquid chromatography–mass spectrometry (HPLC-MS) without derivatization, charge-tagging with HPLC-MS analysis, dimethylglycine derivatization followed by HPLC-ESI-MS and GC–MShigh-performance liquid chromatography/electrospray ionization tandem mass spectrometry (HPLC-ESI-MS) and gas chromatography-mass spectrometry (GC–MS) analysis of oxys-terol trimethylsilyl derivatives), each having peculiar features might account for the variation range in oxysterols levels [27]. We found a significant correlation between the circulating levels of 24-OHC and 25-OHC. In this regard, CYP46, the enzyme mainly responsible for 24-OHC synthesis, was also shown to be capable of synthesizing 25-OHC (ratio: 4:1 (24-OHC:25-OHC)), in cell-based systems [28].

Interestingly, we observed that 27-OHC and 27-OHC/TC plasma levels were significantly higher in males than in females. These findings agree with previously reported data for 27-OHC [29], extending the observation to 27-OHC/TC. Circulating 27-OHC levels are lower in females than in males in both rodents and humans, and this is most likely related to the upregulation of CYP7B1 expression by estradiol and estrogen receptor activation [13,23]. Interestingly, 27-OHC itself has also been regarded has a peculiar selective estrogen receptor modulator (SERM), and has been shown to interfere with the atheroprotective activity of estrogens [30]. These findings may be relevant in the overall context of sex-related ASCVD risk and related response to drugs [31]. Moreover, experimental work indicates that 27-OHC also has an adverse impact on bone mineralization [32] and breast cancer proliferation [33].

In healthy subjects, 24-OHC and 27-OHC were found to correlate with TC, LDL-C and non-HDL-C [24]. In our cohort of moderate hypercholesterolemic patients, we observed a positive trend between plasma 27-OHC and non-HDL-C, but no correlations of oxysterols with TC and LDL-C, although 24-OHC levels positively correlated with apoB concentrations.

In our moderate hypercholesterolemic patients, a significant negative correlation was found between 27-OHC/TC and HDL-C and apoAI, its main lipoprotein component, circulating levels. This observation is in agreement with a previously reported inverse relationship between 27-OHC and HDL-C in normocholesterolemic subjects [34,35]. One possible explanation may be that 27-OHC, acting as liver X receptor (LXR) α ligand, upregulates the expression of cholesteryl ester transfer protein (CETP), which in turn transfers cholesteryl ester from HDL to other lipoproteins, leading to HDL-C reduction [34] and possibly to LDL-C increase [36]. It may be then speculated that the ineffective reverse cholesterol transport in individuals with very low levels of HDL-C may be compensated by this mechanism, which therefore could represent an alternative pathway in the context of the complex regulation of reverse cholesterol transport [37]. In our normo-triglyceridemic population, 24-, 25- and 27-OHC correlated with TG. While the 27-OHC and TG relationship has already been reported in healthy subjects [24], that between 24-, 25-OHC and TG is novel, to the best of our knowledge, and is still present upon normalization by TC (24-, 25-OHC/TC and TG). This correlation could be the consequence of the known LXR modulation by these oxysterols [38] and the resulting stimulation of liver TG synthesis [39].

In the context of the studied population, featuring normal BMI and abdominal circumference, we observed that the 27-OHC/TC ratio positively correlated with abdominal circumference and VFR%. This finding is novel in the clinical setting, although the interrelationship between 27-OHC and adipose tissue has previously been addressed experimentally, leading to apparently controversial observations. In addition to circulate in serum, 27-OHC may also be locally produced by rodent and human adipocytes, where it may counteract adipogenesis [40]. 27-OHC content of white adipose tissue was negatively correlated with adipose mass in mice and exposure to 27-OHC suppressed intracellular TG accumulation by down-regulating lipogenic and adipogenic gene expression during adipocyte maturation of mouse 3T3-L1 cells [41]. However, in mice, 27-OHC administration has been shown to promote adipose tissue hyperplasia, independently from diet type, increasing visceral fat and local inflammation [42]. In light of the relevance of dysfunctional visceral and ectopic adipose for ASCVD [43,44], this intriguing connection requires further clarification.

To our knowledge, no data are available on the effects of hypocholesterolemic nutraceutical treatments on circulating oxysterol levels. Since, in this field, several nutraceutical combinations include RYR, whose main active component is monacolin K, notoriously structurally identical to the statin lovastatin, we may consider a comparison with the available data regarding the impact of statins treatment on 24-, 25- and 27-OHC in subjects with moderate hypercholesterolemia [10,45]. After nutraceutical intervention, 24-OHC level and 24-OHC/TC ratio were unchanged, differently with the reducing effect of simvastatin (80 mg/day) or atorvastatin (40 mg/day) [11,45]. The 25-OHC concentration and 25-OHC/TC ratio were also not affected in both placebo and active groups. Previously reported effects of statins showed decreased 25-OHC concentrations in hypercholesterolemic patients [10,46]. In agreement with the effect of simvastatin (80 mg/day) and atorvastatin (40 mg/day) [11], in our study, 27-OHC levels were significantly reduced in the nutraceutical group, whereas the 27-OHC/TC ratio was not different between arms. 27-OHC levels were also found to be decreased after treatment with atorvastatin or rosuvastatin in subjects with familial hypercholesterolemia or familial combined hyperlipidemia [47]. Due to the similarity of action of RYR extracts and statins, the observed 27-OHC reduction in our study may be the result of cholesterol synthesis inhibition, as also supported by the unchanged 27-OHC/TC ratio. Interestingly, such 27-OHC reduction may also contribute to the observed LDL-C decrease via downregulation of the CETP-pathway. As the nutraceutical combination used here, in addition to RYR, also contains the probiotic *Bifidobacterium longum* BB536, niacin, and coenzyme Q10, one should not exclude some contribution of these components to the effects on the observed reduction in 27-OHC level. One may hypothesize that *Bifidobacterium longum* BB536 may possibly contribute to this reduction by means of its biliary salt hydrolase activity, taking place in the ileum and consequently interfering with the enterohepatic circulation of cholesterol [18].

The evaluation of oxysterol levels after treatment with this nutraceutical combination provides further information on the safety of this product. Oxysterols seem to have a pathogenic role in hyperlipidemia and atherosclerosis both via modulating numerous systemic functions as well as with local actions, and the absolute reduction of 27-OHC concentration may result beneficial, in the context of this cohort of hypercholesterolemic subjects. Elevated circulating 27-OHC may have detrimental effects on the cardiovascular system through multiple mechanisms (SERM activity, LXRα and β ligand), promoting vascular inflammation, which is critically involved in atherogenesis [42,48,49,50]. The additional contribution of 27-OHC and CYP27A1, responsible for its synthesis, which are abundantly present in atherosclerotic plaques, needs further studies [29,51,52].

The findings of the present study may be relevant in terms of long-term safety in consideration of the potentially detrimental role of increased 24-, 25- and 27-OHC also in the context of neurodegenerative diseases such as mild cognitive impairment and dementia [53], as well as bone mineralization and breast cancer [54,55,56].

The main strengths of this study include (1). the extensive exploration of the relationships between 24-, 25- and 27-OHC with a relevant set of CVD risk biomarkers in subjects with moderate hypercholesterolemia, highlighting a series of novel correlations, and (2). the first evaluation of the impact of a nutraceutical combination, designed for the control of hypercholesterolemia, on the circulating levels of these oxysterols. The present study has some limitations. The lack of a control group, useful for comparison, is an intrinsic limitation of the cross-sectional study. Therefore, several observations cannot be extended to healthy subjects. A limitation of the interventional study is also that the dietary intake of the volunteers randomized to either placebo or nutraceutical intervention was not recorded.

Due to their relevant biological effects, the measurement of 24-, 25- and especially 27-OHC may be useful even when assessing the long-term safety of hypocholesterolemic treatments (statins, ezetimibe, bempedoic acid and PCSK9 inhibitors), indicating the need for further larger studies.

## 5. Conclusion

In conclusion, this study adds novel information on this hypocholesterolemic nutraceutical combination, regarding its efficacy and safety, according to oxysterol profile.

## Figures and Tables

**Figure 1 nutrients-13-00427-f001:**
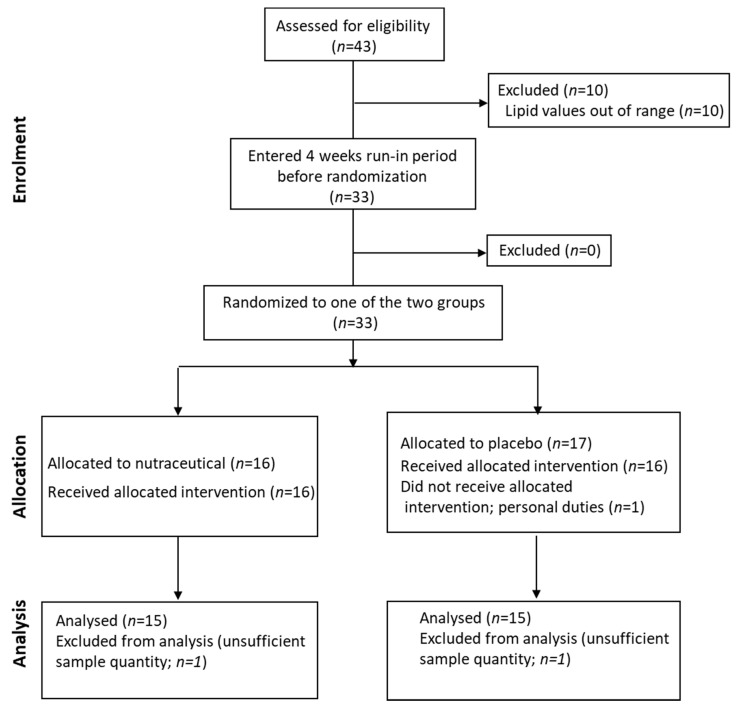
CONSORT statement flow diagram.

**Figure 2 nutrients-13-00427-f002:**
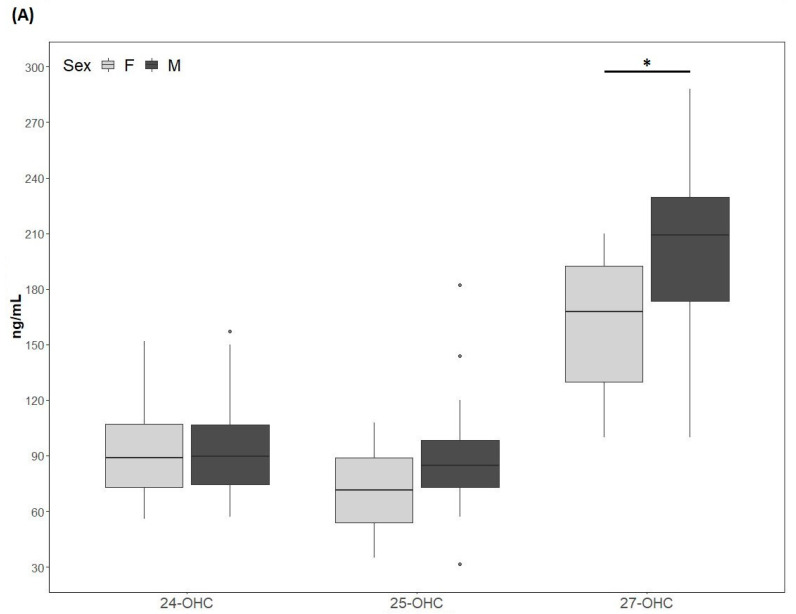
(**A**) 24-hydroxycholesterol (24-OHC), 25-hydroxycholesterol (25-OHC) and 27-hydroxycholesterol (27-OHC) plasma levels; (**B**) 24-OHC/total cholesterol (TC), 25-OHC/TC and 27-OHC/TC. TC values (mean ± SD) were 250.5 ± 75.6 mg/dL for males and 257.9 ± 80 mg/dL for females. M, males; F, females. * *p*-value < 0.01.

**Table 1 nutrients-13-00427-t001:** Main clinical and biochemical characteristics of the whole study population and sorted in the two arms.

	Whole Cohort	Placebo Arm	Nutraceutical Combination Arm	Difference between Arms at Baseline (*p*-Value)
Sex (M/F)	16/14	8/7	8/7	-
Age (years)	57.5 (48, 64)	48 (41, 58)	63 (57, 65)	0.006
Weight (kg)	65 (62, 79)	65 (63, 80)	63 (60, 77)	0.57
BMI (kg/m^2^)	23.84 (21.3, 27.7)	23.58 (20.75, 27.94)	23.9 (22.77, 27.63)	0.71
Abdominal Circumference (cm)	87.25 (83.5, 94)	87.5 (82, 94)	87 (84, 94)	0.75
Waist Circumference (cm)	94.5 (88, 99)	95 (87, 101.5)	94 (90, 99)	0.91
BIA (%)	30.9 (27.2, 38.5)	30.6 (25.5, 39.7)	32.3 (27.3, 38.5)	0.62
VFR (%)	10.5 (7.5, 11.5)	9.5 (6, 13.5)	10.5 (8.5, 11.5)	0.72
SBP (mmHg)	120 (120, 130)	125 (110, 130)	120 (120, 130)	0.53
DBP (mmHg)	80 (80, 80)	80 (80, 80)	80 (80, 80)	0.97
HR (bpm)	64 (60, 68)	65 (64, 68)	64 (60, 68)	0.28
TC (mg/mL)	270 (246, 288)	270 (255, 290)	270 (233, 288)	0.51
LDL-C (mg/mL)	179 (169, 195)	187 (172, 195)	176 (165, 196)	0.33
HDL-C (mg/mL)	56.5 (43, 77)	54 (47, 67)	65 (42, 83)	0.67
non-HDL-C (mg/mL)	209.5 (192, 230)	214 (196, 234)	207 (188, 216)	0.31
TG (mg/mL)	114.5 (95, 153)	112 (92, 130)	127 (95, 159)	0.57
apoAI (mg/dL)	114.5 (95, 132)	113 (95, 129)	125 (91, 141)	0.39
apoB (mg/dL)	146 (135, 155)	143 (133, 146)	155 (142, 158)	0.03
oxLDL (U/L)	76.6 (70, 85.2)	71.7 (67.3, 84.6)	76.8 (74.5, 123.8)	0.17
24-OHC (ng/mL)	89 (73, 109)	89 (73, 103)	90.9 (71.8, 110)	0.72
24-OHC/TC (ng/mg)	34 (27, 41)	33 (27, 39)	35 (26, 43)	0.71
25-OHC (ng/mL)	84.2 (60.5, 96)	86 (63, 106)	81 (57, 96)	0.55
25-OHC/TC (ng/mg)	29 (25, 38)	31 (25, 39)	28 (22, 38)	0.87
27-OHC (ng/mL)	183.5 (152, 211)	174 (115, 219)	190 (166.3, 211)	0.6
27-OHC/TC (ng/mg)	72 (51, 80)	72 (47, 74)	73 (51, 88)	0.3
Lp(a) (mg/dL)	6 (4, 11)	4 (2, 9)	7 (5, 13)	0.09
PCSK9 (ng/dL)	339.87 (283.17, 403.96)	340.04 (279.63, 402.52)	339.7 (283.17, 410.09)	0.89
FPG (mg/dL)	93.5 (89, 97)	95 (89, 97)	92 (89, 103)	0.98
Insulin (mUI/L)	3.38 (2.42, 5.04)	3.08 (2.49, 6.2)	3.38 (2.13, 5.04)	0.77
HOMA-IR	0.75 (0.56, 1.18)	0.72 (0.57, 1.44)	0.77 (0.47, 1.18)	0.81
FGF19 (pg/mL)	222.87 (173.05, 330.6)	232.4 (106.58, 333.98)	215.61 (176.79, 237.12)	0.94
FGF21 (pg/mL)	174.86 (118.32, 237.14)	157.08 (71.66, 226.57)	178.62 (158.16, 364.15)	0.25
Creatinine (mg/dL)	0.8 (0.7, 0.9)	0.8 (0.7, 1)	0.8 (0.7, 0.9)	0.76

Data are median (Q1, Q3). BMI: body mass index, BIA: bioelectrical impedance analysis/abdominal fat mass, VFR: visceral fat rating, SBP: systolic blood pressure, DBP: diastolic blood pressure, HR: heart rate, TC: total cholesterol, LDL-C: low-density lipoprotein. cholesterol, HDL-C: high-density lipoprotein cholesterol, TG: triglycerides, apoAI: apolipoprotein AI, apoB: apolipoprotein B, oxLDL: oxidized LDL, 24-OHC: 24-hydroxycholesterol, 25-OHC: 25-hydroxycholesterol, 27-OHC: 27-hydroxycholesterol, Lp(a): lipoprotein (a), PCSK9: proprotein convertase subtilisin/kexin type 9, FPG: fasting plasma glucose, HOMA-IR: Homeostatic Model Assessment of Insulin Resistance, FGF: fibroblast growth factor.

**Table 2 nutrients-13-00427-t002:** Correlation of circulating oxysterols levels and total cholesterol-normalized circulating oxysterols levels with clinical and biochemical characteristics of the study population.

	24-OHC	24-OHC/TC	25-OHC	25-OHC/TC	27-OHC	27-OHC/TC
Sex	0.68	0.42	0.14	0.07	0.02	0.008
Age	−0.269/0.151	−0.279/0.135	−0.29/0.121	−0.285/0.127	0.191/0.313	0.218/0.248
Weight	0.025/0.895	0.158/0.403	0.133/0.483	0.225/0.232	0.236/0.209	0.339/0.067
BMI	0.069/0.728	0.198/0.313	0.124/0.528	0.23/0.238	0.148/0.453	0.265/0.173
Abdominal Circumference	−0.034/0.859	0.118/0.536	0.06/0.754	0.159/0.401	0.272/0.146	0.414/0.023
Waist Circumference	0.037/0.847	0.109/0.567	−0.003/0.988	0.04/0.833	0.042/0.824	0.104/0.586
BIA	−0.054/0.776	0.06/0.753	−0.253/0.178	−0.171/0.367	0.026/0.89	0.136/0.475
VFR	0.021/0.912	0.168/0.376	0.138/0.468	0.253/0.177	0.292/0.118	0.42/0.021
SBP	−0.051/0.789	−0.094/0.62	0.172/0.365	0.182/0.336	−0.11/0.561	−0.135/0.477
DBP	−0.132/0.485	−0.194/0.304	−0.16/0.397	−0.175/0.354	−0.126/0.505	−0.18/0.34
HR	0.095/0.617	−0.012/0.95	0.128/0.499	0.073/0.7	−0.002/0.992	−0.105/0.569
TC	0.219/0.246	−0.193/0.307	0.23/0.221	−0.088/0.643	0.158/0.405	−0.295/0.115
LDL-C	0.022/0.909	−0.314/0.091	0.12/0.528	−0.151/0.426	0.23/0.221	−0.148/0.435
HDL-C	0.066/0.729	−0.153/0.419	−0.033/0.864	−0.188/0.319	−0.252/0.179	−0.489/0.006
non-HDL-C	0.215/0.253	−0.131/0.49	0.29/0.12	0.013/0.946	0.341/0.065	−0.041/0.828
TG	0.517/0.004	0.41/0.024	0.481/0.007	0.401/0.028	0.353/0.056	0.253/0.178
apoAI	0.104/0.586	−0.183/0.333	0.04/0.833	−0.164/0.387	−0.052/0.783	−0.361/0.05
apoB	0.455/0.012	0.306/0.1	0.155/0.415	0.025/0.897	0.201/0.287	0.032/0.866
oxLDL	0.202/0.285	0.155/0.414	−0.094/0.62	−0.136/0.472	−0.04/0.835	−0.045/0.815
24-OHC	-	0.912/<0.0001	0.626/0.00021	0.552/0.0015	−0.084/0.655	−0.177/0.347
24-OHC/TC	0.912/<0.0001	-	0.518/0.0033	0.580/0.00078	−0.145/0.443	−0.049/0.793
25-OHC	0.626/0.00021	0.518/0.0033	-	0.945/<0.0001	0.053/0.779	−0.060/0.749
25-OHC/TC	0.552/0.0015	0.580/0.00078	0.945/<0.0001	-	−0.0001/0.999	0.032/0.865
27-OHC	−0.084/0.655	−0.145/0.443	−0.053/0.779	−0.0001/0.999	-	0.892/<0.0001
27-OHC/TC	−0.117/0.347	−0.049/0.793	−0.06/0.749	0.032/0.865	0.892/<0.0001	-
Lp(a)	0.324/0.081	0.419/0.021	0.369/0.045	0.448/0.013	0.084/0.658	0.168/0.374
PCSK9	0.035/0.852	−0.188/0.321	0.067/0.724	−0.107/0.574	−0.223/0.236	−0.45/0.013
FPG	0.1/0.598	0.097/0.609	−0.015/0.938	−0.023/0.905	0.227/0.228	0.247/0.189
Insulin	0.019/0.92	0.92/0.419	0.073/0.7	0.176/0.352	0.14/0.46	0.272/0.146
HOMA-IR	0.029/0.879	0.153/0.42	0.085/0.654	0.181/0.339	0.166/0.38	0.292/0.117
FGF19	−0.099/0.637	−0.003/0.987	0.133/0.528	0.231/0.267	−0.225/0.279	−0.118/0.573
FGF21	−0.178/0.454	−0.208/0.379	0.218/0.357	0.197/0.405	0.013/0.956	0.043/0.857
Creatinine	−0.064/0.738	0.069/0.716	0.101/0.597	0.218/0.247	0.433/0.017	0.567/0.001

Pearson correlation coefficient and *P*-value are reported for each correlation, except for Sex (dichotomic variable). BMI: body mass index, BIA: bioelectrical impedance analysis/abdominal fat mass, VFR: visceral fat rating, SBP: systolic blood pressure, DBP: diastolic blood pressure, HR: heart rate, TC: total cholesterol, LDL-C: low-density lipoprotein. cholesterol, HDL-C: high-density lipoprotein cholesterol, TG: triglycerides, apoAI: apolipoprotein AI, apoB: apolipoprotein B, oxLDL: oxidized LDL, 24-OHC: 24-hydroxycholesterol, 25-OHC: 25-hydroxycholesterol, 27-OHC: 27-hydroxycholesterol, Lp(a): lipoprotein (a), PCSK9: proprotein convertase subtilisin/kexin type 9, FPG: fasting plasma glucose, HOMA-IR: Homeostatic Model Assessment of Insulin Resistance, FGF: fibroblast growth factor.

**Table 3 nutrients-13-00427-t003:** Determination of serum levels of 24-hydroxycholesterol (24-OHC), 25-hydroxycholesterol (25-OHC) and 27-hydroxycholesterol (27-OHC) and their ratio to total cholesterol (TC).

	Placebo		Nutraceutical		Difference of Changes between Arms
	Baseline	12 Weeks	Baseline	12 Weeks	*p*-Value
24-OHC (ng/mL)	89 (73, 103)	97 (87, 118)	91 (72, 110)	94 (75, 139)	0.2
25-OHC (ng/mL)	86 (63, 106)	79 (52, 96)	81 (57, 96)	81 (62, 91)	0.91
27-OHC (ng/mL)	174 (115, 219)	179 (111, 232)	190 (166, 211)	170 (133, 187)	0.03
24-OHC/TC (ng/mg)	33 (27, 39)	34 (30, 41)	35 (26, 43)	40 (35, 51)	0.57
25-OHC/TC (ng/mg)	31 (25, 39)	25 (20, 34)	28 (22, 38)	35 (27, 41)	0.36
27-OHC/TC (ng/mg)	72 (47, 74)	67 (43, 82)	73 (51, 88)	74 (54, 94)	0.09

Data are shown as median (1st quartile, 3rd quartile), *p*-values are adjusted for age and apolipoprotein B.

## Data Availability

The data presented in this study are available on request from the corresponding author. The data are not publicly available due to privacy reasons.

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
