# Peer review of "Interactions of Oxysterols with Atherosclerosis Biomarkers in Subjects with Moderate Hypercholesterolemia and Effects of a Nutraceutical Combination (Bifidobacterium longum BB536, Red Yeast Rice Extract) (Randomized, Double-Blind, Placebo-Controlled Study)"

_nutrients, 2021, doi:10.3390/nu13020427_

Round 1
Reviewer 1 Report
The authors report an interesting study primarily focused on the interaction between oxysterols and atherosclerosis biomarkers in individuals with moderate hypercholesterolaemia in the context of primary prevention. Furthermore, the study also evaluated a nutraceutical approach (Bifidobacterium longum BB536 + Red yeast rice extract + niacin + coenzyme Q10). Overall, the authors report mainly relationships between oxysterols (24-OHC, 25-OHC and 27-OHC) and CV risk biomarkers, as the nutraceutical approach only reduced 27-OHC levels. Based on the results, the authors claim that these findings support the safety and efficacy of this nutraceutical approach.
Major issues that must be addressed before the manuscript is considered:
1. The first major issue is the claim that "the findings (...) support the safety and efficacy of this nutraceutical approach" (L. 42). In reality, the intervention produced very limited efficacy, specifically on 27-OHC levels. No effect was seen on other atherosclerosis biomarkers and oxysterols. Based on this, the authors' claim seems to be an overstatement of the real effect. Also, methodological issues that are described below limit the interpretation of the findings.
2. Outcome measures: The manuscript lacks clear definition and transparency in the reporting of outcome measures for efficacy and safety (to support the authors claim in the statement above). In fact, the authors only report changes in TC and oxysterols (subsection 3.4). However, the ClinicalTrials.gov record for this trial (NCT02689934) includes outcome measures which, intringuingly, were not explored or defined in the manuscript, as follows:
2.1. Primary outcome: Percent change in LDL-C from baseline (at 12 weeks)
2.2. Secondary outcomes: (a) safety and tolerability (incidence of adverse effects), (b) changes in other lipids (HDL-C, non HDL-C, TG and apolipoproteins), (c) changes in vital signs
3. Inclusion criteria (L. 85): this study included subjects in primary prevention for CVD, with an age between 18 to 70 years, LDL-C levels between 130-200 mg/dL and non-smokers. The main question here is what defines "subjects in primary prevention". Were the subjects on some type of intervention, e.g. statin medication or diet/lifestyle modification?
4. An exclusion criteria was untreated arterial hypertension (L. 88). It seems this refers to treated vs non-treated with medication (L. 103). However, a hypertensive patient could only be following lifestyle modifications (no information available on this). In fact, a better distinction would be provided by the dichotomy of controlled vs uncontrolled hypertension. Interestingly, the ClinicalTrials.gov record for this trial (NCT02689934) states uncontrolled or severe hypertension as an exclusion criteria. The authors need to clarify this exclusion criteria. Also, more attention should have be given to this major CV risk factor throughout the manuscript, and the prevalence of hypertension should be provided for both groups.
5. Study design and population: 1 subject in each group was excluded from the analysis (Figure 1). The reasons for the exclusion must be specified.
6. Another major issue is the fact that the trial arms (intervention vs placebo) are not similar in terms of baseline characteristics (Table 1). In fact, age and apoB were different between the 2 groups (L. 206-207). However, the authors have not provided the magnitude of the difference nor the median values for each group. Although the authors state that P-values for "Difference of changes between arms" (Table 2) were adjusted for age, there is no information available on how the authors controlled the effect for age. Also, the results were not controlled for differences in apoB, which is an important factor in atherosclerotic CV disease. Therefore, the interpretation of the findings is largely hindered.
7. After intervention, TC is the only outcome measure that was mentioned (L. 207-208) besides oxysterols (Table 2). For a more complete picture, the authors should have included other relevant parameters, namely lipid levels (e.g. LDL-C, oxLDL and others) as mentioned above.
Additional issues:
1. In the abstract, results (L. 34) should be presented in a more complete way rather than only say "were (significantly) correlated" or "was higher in men vs. women", for instance. Mean differences or absolute values for each group should be included to provide a clear understanding of the magnitude of the observed differences.
2. Exclusion criteria: one of the criteria is "any pharmacological treatments known to interfere with the study treatment". For the purpose of complete transparency in reporting, these should be specified.
3. There are discrepancies between the values presented in the text (L. 156-157) vs Table 1:
3.1. "Median TC was 271 (247, 288) mg/dL" (L. 156) vs "270 (246, 288) mg/dL" (Table 1)
3.2. "LDL-C was 180 (170, 196) mg/dL" (L. 156-157) vs "179 (169, 195) mg/dL" (Table 1)
4. Table 1 (L. 180):
4.1. The table only presents median values for the cohort. Values for intervention and placebo groups must also be presented, rather than presenting only the P-value for the difference, which therefore is not known based on the presented data.
4.2. Correlations are presented as footnote to the table. This information should be presented in other ways, e.g. independent table or graphs, and correlation coeficients should also be included in order to distinguish mild, moderate and strong correlations.
5. Figure 2 (L. 200):
5.1. Where is graph (B)?
5.2. Also, boxplots are not completely suitable for this kind of data. It would be better to use scatterplots with median bar.
Minor points:
L. 93-95 - "((Lactoforene Colesterolo (...) n=16)", check for the parenthesis
L. 109-110 - "The present analysis (...) circulating oxysterol levels", check the phrasing
L. 125-126 - "HP5 (Hewlett Packard, USA)", is the manufacturer correct?
L. 251 - "mild hypercholesterolemic patients", correct mild to moderate, in order to match the rest of the manuscript
Finally, the hypercholesterolaemia/hypercholesterolemia terminology must be revised throughout the manuscript.
Author Response
The authors report an interesting study primarily focused on the interaction between oxysterols and atherosclerosis biomarkers in individuals with moderate hypercholesterolaemia in the context of primary prevention. Furthermore, the study also evaluated a nutraceutical approach (Bifidobacterium longum BB536 + Red yeast rice extract + niacin + coenzyme Q10). Overall, the authors report mainly relationships between oxysterols (24-OHC, 25-OHC and 27-OHC) and CV risk biomarkers, as the nutraceutical approach only reduced 27-OHC levels. Based on the results, the authors claim that these findings support the safety and efficacy of this nutraceutical approach.
Thanks for your positive comments.
Major issues that must be addressed before the manuscript is considered:
- The first major issue is the claim that "the findings (...) support the safety and efficacy of this nutraceutical approach" (L. 42). In reality, the intervention produced very limited efficacy, specifically on 27-OHC levels. No effect was seen on other atherosclerosis biomarkers and oxysterols. Based on this, the authors' claim seems to be an overstatement of the real effect. Also, methodological issues that are described below limit the interpretation of the findings.
Thanks, according to your comment, we mitigated this comment and the sentence was modified as follows (lines 43-44): “In addition, no adverse changes of OHCs levels upon nutraceutical treatment were found.”
- Outcome measures: The manuscript lacks clear definition and transparency in the reporting of outcome measures for efficacy and safety (to support the authors claim in the statement above). In fact, the authors only report changes in TC and oxysterols (subsection 3.4). However, the ClinicalTrials.gov record for this trial (NCT02689934) includes outcome measures which, intringuingly, were not explored or defined in the manuscript, as follows:
2.1. Primary outcome: Percent change in LDL-C from baseline (at 12 weeks)
2.2. Secondary outcomes: (a) safety and tolerability (incidence of adverse effects), (b) changes in other lipids (HDL-C, non HDL-C, TG and apolipoproteins), (c) changes in vital signs
Thanks for this important comment. As indicated in the answer to comment 1, the statement on efficacy and safety has been mitigated. Moreover, at the end of the Introduction section, we refer to the previous study (ref. 18) that addressed the above-reported outcomes, by adding, at lines 77-79, the sentence: “This nutraceutical combination was previously found to be quite effective in reducing LDL cholesterol (LDL-C) and total cholesterol (TC) levels in moderately hypercholesterolemic subjects [18].” Moreover, the effect of this nutraceutical on LDL-C levels was also added in now section 3.3 (also former 3.4), line 279.
- Inclusion criteria (L. 85): this study included subjects in primary prevention for CVD, with an age between 18 to 70 years, LDL-C levels between 130-200 mg/dL and non-smokers. The main question here is what defines "subjects in primary prevention". Were the subjects on some type of intervention, e.g. statin medication or diet/lifestyle modification?
Thank you. “Subjects in primary (cardiovascular) prevention” are defined as subjects with no prior cardiocerebrovascular disease. The study cohort included some subjects (15.6%) undergoing drug therapy for arterial hypertension. This has been reported now at lines 95-97, as follows: “The study cohort included some subjects (15.6%) undergoing drug therapy for arterial hypertension, as reported in [18].”
- An exclusion criteria was untreated arterial hypertension (L. 88). It seems this refers to treated vs non-treated with medication (L. 103). However, a hypertensive patient could only be following lifestyle modifications (no information available on this). In fact, a better distinction would be provided by the dichotomy of controlled vs uncontrolled hypertension. Interestingly, the ClinicalTrials.gov record for this trial (NCT02689934) states uncontrolled or severe hypertension as an exclusion criteria. The authors need to clarify this exclusion criteria. Also, more attention should have be given to this major CV risk factor throughout the manuscript, and the prevalence of hypertension should be provided for both groups.
Thanks for this specific comment and sorry for this omission. In line with the ClinicalTrials.gov record for this trial (NCT02689934), lines 92-93 now read: “with untreated, uncontrolled or severe arterial hypertension;”. As suggested, in lines 104-105, the following sentence was added “The prevalence of subjects with drug-controlled hypertension was 18.8% in the placebo arm and 12.5% in the intervention arm.”.
- Study design and population: 1 subject in each group was excluded from the analysis (Figure 1). The reasons for the exclusion must be specified.
Thanks, 1 sample per treatment arm was quantitatively not sufficient for the required analyses and therefore the case was excluded. This observation has been added to Figure 1.
- Another major issue is the fact that the trial arms (intervention vs placebo) are not similar in terms of baseline characteristics (Table 1). In fact, age and apoB were different between the 2 groups (L. 206-207). However, the authors have not provided the magnitude of the difference nor the median values for each group. Although the authors state that P-values for "Difference of changes between arms" (Table 2) were adjusted for age, there is no information available on how the authors controlled the effect for age. Also, the results were not controlled for differences in apoB, which is an important factor in atherosclerotic CV disease. Therefore, the interpretation of the findings is largely hindered.
Thanks for this important observation. The median/quartile data relative to the nutraceutical intervention are now shown in the new Table 3, reporting the values relative to each randomization arm, plus the P value. When we have heterogeneity in experimental units sometimes restrictions on the randomization (blocking) can improve the test for treatment effects. In some cases, we don’t have the opportunity to construct blocks, but can recognize and measure a continuous variable as contributing to the heterogeneity in the experimental units. We performed a mixed linear model. The mixed linear model provides the flexibility of modeling not only the means of the data but their variances and covariances as well. Specifically, age as well as (now) ApoB have been added in the model as covariate in order to adjust the main evaluation. Thus, the footnote to Table 4 now reads “Data are shown as median (1st quartile, 3rd quartile), P-values are adjusted for age and apolipoprotein B….”.
- After intervention, TC is the only outcome measure that was mentioned (L. 207-208) besides oxysterols (Table 2). For a more complete picture, the authors should have included other relevant parameters, namely lipid levels (e.g. LDL-C, oxLDL and others) as mentioned above.
Thanks, LDL-C levels after intervention was mentioned in line 280. Additional information may be found in ref. 18. In this regard, the following words have been added in line 281: “(please refer to this article for additional data on CVD biomarkers)”.
Additional issues:
- In the abstract, results (L. 34) should be presented in a more complete way rather than only say "were (significantly) correlated" or "was higher in men vs. women", for instance. Mean differences or absolute values for each group should be included to provide a clear understanding of the magnitude of the observed differences.
Thanks for this relevant comment. The Results section in the Abstract has been modified accordingly, as follows: Lines 35-37: “27-OHC and 27-OHC/total cholesterol (TC) were higher in men (median 209 ng/mL and 77 ng/mg, respectively) vs. women (median 168 ng/mL and 56 ng/mg, respectively)”. Moreover, now line 40 reads: “After intervention, 27-OHC levels were significantly reduced by 10.4% in the…”.
- Exclusion criteria: one of the criteria is "any pharmacological treatments known to interfere with the study treatment". For the purpose of complete transparency in reporting, these should be specified.
Thank you. The following words were added in line 94: “(including statins, ezetimibe, fibrates, thyroid hormones)”.
- There are discrepancies between the values presented in the text (L. 156-157) vs Table 1:
3.1. "Median TC was 271 (247, 288) mg/dL" (L. 156) vs "270 (246, 288) mg/dL" (Table 1)
3.2. "LDL-C was 180 (170, 196) mg/dL" (L. 156-157) vs "179 (169, 195) mg/dL" (Table 1)
Thanks. Sorry for these mistakes. The values have now been corrected in the text according to the correct values, which have been reported in Table 1.
- Table 1 (L. 180):
4.1. The table only presents median values for the cohort. Values for intervention and placebo groups must also be presented, rather than presenting only the P-value for the difference, which therefore is not known based on the presented data.
Thank you for this relevant point. Table 1 now reports median values of the whole cohort. A new table (Table 3) presenting the baseline values for intervention and placebo groups and the related P-value for the difference has been added.
4.2. Correlations are presented as footnote to the table. This information should be presented in other ways, e.g. independent table or graphs, and correlation coefficients should also be included in order to distinguish mild, moderate and strong correlations.
Thanks, the correlations are now shown in a separate table (new Table 2) which includes correlation coefficients and P-values.
- Figure 2 (L. 200):
5.1. Where is graph (B)?
Graph (B) was included in the first submission, but for some reasons was then omitted from the version you received. Graph 2B has been now added again.
5.2. Also, boxplots are not completely suitable for this kind of data. It would be better to use scatterplots with median bar.
Thanks for this comment. In figure 2 we wanted to evaluate and show the relationship with sex, a dichotomous variable. In this case a box plot could be a simple way of summarizing this set of data measured on an interval scale. The use of a scatter plot may be useful to determine if there are patterns or correlations between two variables. A scatter plot may be used when an independent variable has multiple values for the dependent variable. For these reasons we propose the box plots.
Minor points:
- 93-95 - "((Lactoforene Colesterolo (...) n=16)", check for the parenthesis
Done
- 109-110 - "The present analysis (...) circulating oxysterol levels", check the phrasing
Thanks, current lines 116-117 now read “In the present analysis, based upon the study reported in [18], we evaluated basal and post-intervention circulating oxysterol levels.”.
- 125-126 - "HP5 (Hewlett Packard, USA)", is the manufacturer correct?
Sorry, in line 133 the manufacturer was corrected to “Agilent”.
- 251 - "mild hypercholesterolemic patients", correct mild to moderate, in order to match the rest of the manuscript
Thank you, done in current line 331.
Finally, the hypercholesterolaemia/hypercholesterolemia terminology must be revised throughout the manuscript.
Sorry for this. The terminology has been revised throughout the manuscript and the word “hypercholesterolemia” was chosen.
Reviewer 2 Report
The manuscript describes a randomized, double-blind, placebo-controlled dietary intervention study aiming to investigate whether a supplement containing Bifidobacterium longum, red yeast rice extract, niacin and Q10 can modulate serum oxysterol levels in a cohort of mildly hypercholesterolemic subjects. A secondary aim of the study is to investigate correlations between the oxysterol levels with atherosclerosis risk factors. Actually, the main results of this intervention were presented in Ref.19 and the novelty of this study lies mainly on the measurement of oxysterols.
Main comments
2.1. Study design and population: A major shortcoming of the study is that the investigators didn’t record the dietary intakes of the volunteers before and after the intervention. It should be inserted as a limitation in the discussion section
2.3. Biochemical and immunometric assays: Please define the standard automated clinical procedure
2.5. Statistical analysis: A power analysis should be presented
2.5. Statistical analysis: The results are presented as median and interquartile ranges implying that the data do not follow a normal distribution and generally the investigators used non-parametric test for their analyses. So why did they choose to use the Pearson correlation instead of Spearman for example.
3. Results: Renumber the subsections of the Results. Paragraph 3.3 should be 3.2 and so on
3.1. Study population and Table 1: I don’t understand why did the authors choose to present the baseline data in the format of Table 1. Even if there are not many significant differences between the groups, I believe that Table 1 would be more informative if the baseline data were presented per group.
Table 1. Please identify in the footnote that BIA (%) stands for abdominal fat mass (%)
Table 1. The correlation analysis should be presented in a separate Table and the main significant correlations should be presented as scatter dot diagrams, too. Moreover, the Pearson’s r coefficient should be also presented. In any case I believe that a linear regression analysis is more powerful in showing associations between variables in the presence of covariates.
4. Discussion: A paragraph describing the strengths and limtations of the study should be added in the discussion section
4. Discussion: The authors try to explain the effect of the nutraceutical on oxysterol levels based on the presence of monakolin in the RYR extract. However, the nutraceutical also contains other bioactive ingredients which are ignored in the discussion section. Do the authors believe that only monakolin in responsible for the observed changes in 27-OHC levels after the intervention ?
4. Discussion: My main concern regarding the interpretation of the Results is that the authors try to explain the differences of oxysterols in serum by possible changes observed intracellularly. Are the extracellular levels of oxysterols determined solely by the intracellular metabolism of cholesterol ?
Author Response
The manuscript describes a randomized, double-blind, placebo-controlled dietary intervention study aiming to investigate whether a supplement containing Bifidobacterium longum, red yeast rice extract, niacin and Q10 can modulate serum oxysterol levels in a cohort of mildly hypercholesterolemic subjects. A secondary aim of the study is to investigate correlations between the oxysterol levels with atherosclerosis risk factors. Actually, the main results of this intervention were presented in Ref.19 and the novelty of this study lies mainly on the measurement of oxysterols.
Main comments
2.1. Study design and population: A major shortcoming of the study is that the investigators didn’t record the dietary intakes of the volunteers before and after the intervention. It should be inserted as a limitation in the discussion section.
Thanks, this has been done. See lines 401-402, reporting the following text: “A limitation of the interventional study is that the dietary intake of the volunteers randomized to either placebo or nutraceutical intervention was not recorded.”.
2.3. Biochemical and immunometric assays: Please define the standard automated clinical procedure
Thanks, we indicated in lines 124-125 the specific diagnostic system (Cobas system, Roche, Italy)
2.5. Statistical analysis: A power analysis should be presented
Thank you. In section 2.5, we added a paragraph reporting the power analysis done on the original primary outcome (LDL-C level change).
2.5. Statistical analysis: The results are presented as median and interquartile ranges implying that the data do not follow a normal distribution and generally the investigators used non-parametric test for their analyses. So why did they choose to use the Pearson correlation instead of Spearman for example.
Thank you for this relevant comment. Taken into account the rather small sample size of this cohort, it becomes difficult to estimate whether there is a non-linear correlation between the factors considered, for example using Spearman correlation to test monotonic relationships (whether linear or not). As this is mostly an explorative analysis, we decided to test whether or not behind these relationships there was a linear association, using Pearson correlations. Thus, we focused our analysis on assessing linear relationships, also taking into account the rather small cohort (n=30). For the same reason, performing linear regression models also seems to be underpowered.
- Results: Renumber the subsections of the Results. Paragraph 3.3 should be 3.2 and so on
Thanks and sorry for this. Done.
3.1. Study population and Table 1: I don’t understand why did the authors choose to present the baseline data in the format of Table 1. Even if there are not many significant differences between the groups, I believe that Table 1 would be more informative if the baseline data were presented per group.
Thank you for this relevant point. Table 1 now reports median values of the whole cohort. A new table (Table 3) presenting the baseline values for intervention and placebo groups and the related P-value for the difference has been added.
Table 1. Please identify in the footnote that BIA (%) stands for abdominal fat mass (%)
Thanks, done.
Table 1. The correlation analysis should be presented in a separate Table and the main significant correlations should be presented as scatter dot diagrams, too. Moreover, the Pearson’s r coefficient should be also presented. In any case I believe that a linear regression analysis is more powerful in showing associations between variables in the presence of covariates.
Thanks. Correlations are now presented in a separate Table (new Table 2), which also includes the Pearson’s r coefficient. Given the many correlation analyses carried out, we prefer to present these results in a table rather than to add few or many figures, potentially selected without a specific rationale. As indicated above (reply to point 2.5), taken into account the low sample size of this cohort, it becomes difficult to estimate whether there is a non-linear correlation between the factors considered, for example using Spearman correlation to test monotonic relationships (whether linear or not). As this is mostly an explorative analysis, we decided to test whether or not behind these relationships there was a linear association, using Pearson correlations. Thus, we focused our analysis on assessing linear relationships, also taking into account the rather small cohort (n=30). For the same reason, performing linear regression models also seems to be underpowered.
- Discussion: A paragraph describing the strengths and limitations of the study should be added in the discussion section
Thanks for this important point. At lines 392-400, the following paragraph has been added: “The main strengths of this study include 1. the extensive exploration of the relationships between 24-, 25- and 27-OHC with a relevant set of CVD risk biomarkers in subjects with moderate hypercholesterolemia, highlighting a series of novel correlations, and 2. the first evaluation of the impact of a nutraceutical combination, designed for the control of hypercholesterolemia, on the circulating levels of these oxysterols. The present study has some limitations. The lack of a control group, useful for comparison, is an intrinsic limitation of the cross-sectional study. Therefore, several observations cannot be extended to healthy subjects. A limitation of the interventional study is that the dietary intake of the volunteers randomized to either placebo or nutraceutical intervention was not recorded.”.
- Discussion: The authors try to explain the effect of the nutraceutical on oxysterol levels based on the presence of monakolin in the RYR extract. However, the nutraceutical also contains other bioactive ingredients which are ignored in the discussion section. Do the authors believe that only monakolin in responsible for the observed changes in 27-OHC levels after the intervention ?
Thanks for this very important comment. Although monacolin k/RYR seem to be the main component responsible for the observed changes, among the other components, the probiotic could specifically play some role.
Starting at line 374, this paragraph was thus added: “As the nutraceutical combination used here, in addition to RYR, also contains the probiotic Bifidobacterium longum BB536, niacin, and coenzyme Q10, one should not exclude some contribution of these components to the effects on the observed reduction in 27-OHC level. One may hypothesize that Bifidobacterium longum BB536 may possibly contribute to this reduction by means of its biliary salt hydrolase activity, taking place in the ileum and consequently interfering with the enterohepatic circulation of cholesterol [18].”.
- Discussion: My main concern regarding the interpretation of the Results is that the authors try to explain the differences of oxysterols in serum by possible changes observed intracellularly. Are the extracellular levels of oxysterols determined solely by the intracellular metabolism of cholesterol ?
Thanks, the circulating levels of oxysterols, as steady-state levels, are determined by intracellular metabolism of cholesterol as well as by clearance from serum by excretory organs. This last aspect has been addressed also by adding the paragraph indicated just above.
Round 2
Reviewer 1 Report
Raised issues have been generally addressed.
Table 2 has several values in bold, but it is not clear what does this mean as not all these values represent significant correlations.
Table 1 and 3 could be merged into one single table which countains the full cohort, the placebo and treatment arms and statistical significance of differences between arms.
Author Response
Thanks to Reviewer 1 for the additional suggestions, which have been fully implemented, as follows.
Table 2 has several values in bold, but it is not clear what does this mean as not all these values represent significant correlations.
Thanks, for the sake of clarity, all bold highlights have been remoed.
Table 1 and 3 could be merged into one single table which countains the full cohort, the placebo and treatment arms and statistical significance of differences between arms.
Thanks, table 1 and 3 have been merged in new table 1.
Reviewer 2 Report
The authors responded adequately to my comments and the presentation and scientific quality of the paper were improved.
Author Response
Thank you.